# Common mental health and emotional and behavioural disorders among adolescents and young adults in Harare and Mashonaland East, Zimbabwe: a population-based prevalence study

Aoife Margaret Doyle [1,2] T Bandason,[2] E Dauya,[2] Grace McHugh,[2] Chris Grundy,[1] Victoria Simms [1,2] D Chibanda,[3,4] Rashida Ferrand[2,5]

For numbered affiliations see end of article.

**Correspondence to**
Aoife Margaret Doyle;
Aoife.Doyle@lshtm.ac.uk

## ABSTRACT

**Objectives** To estimate the prevalence of common mental health disorders (CMDs) and emotional and behavioural disorders among young people and to explore the correlates of CMDs risk.

**Setting** Five urban and periurban communities in Harare and Mashonaland East, Zimbabwe

**Design** Population-based cross-sectional study

**Participants** Young people aged 13–24 years living in households in the study areas.

**Outcome measures** The primary outcome was the proportion of participants screening positive for probable CMDs defined as a Shona Symptoms Questionnaire (SSQ) score ≥8. Secondary outcomes were emotional and behavioural disorders measured using the Strength and Difficulties Questionnaire (SDQ), and adjusted ORs for factors associated with CMD.

**Results** Out of 634 young people, 37.4% (95% CI 33.0% to 42.0%) screened positive for probable CMDs, 9.8% (95% CI 7.5% to 12.7%) reported perceptual symptoms and 11.2% (95% CI 9.0% to 13.8%) reported suicidal ideation. Using UK norms to define normal, borderline and abnormal scores for each of the SDQ domains, a high proportion (15.8%) of Zimbabwean young people had abnormal scores for emotional symptoms and a low proportion had abnormal scores for hyperactivity/inattention scores (2.8%) and prosocial scores (7.1%). We created local cut-offs for the emotional symptoms, hyperactivity/attention and prosocial SDQ domains. The odds of probable CMDs increased with each year of age (OR 1.09, p<0.001) and was higher among those who were out of school and not working compared with those in school or working (adj. OR 1.67 (1.07, 2.62), p=0.04). One in five participants (22.1%) were referred immediately for further clinical assessment but uptake of referral services was low.

**Conclusions** We observed a high prevalence of symptoms of CMDs among general population urban and peri-urban young people especially among those with no employment. There is a need for more accessible and acceptable youth-friendly mental health services.

## STRENGTHS AND LIMITATIONS OF THIS STUDY

⇒ In the context of limited population-level data on the mental health of young people, we included a representative general population sample of young people in five urban and peri-urban communities in Zimbabwe.

⇒ Demonstration of the feasibility of use of audio-computer assisted self-interview to collect self-reported measures of mental health and emotional and behavioural disorders in this young population.

⇒ The Shona Symptoms Questionnaire and Strength and Difficulties Questionnaire (SDQ) tools have not been extensively validated for use in the full age group and the translation of the SDQ into Shona did not follow recommended procedures. Additional validation studies are needed for tools to measure adolescent mental health.

## BACKGROUND

Mental health disorders such as depressive, anxiety and childhood behavioural disorders are common among young people (YP) globally,[1] and in 2019 were among the top causes of morbidity among 10–19-year olds.[2] Poor mental health among adolescents is associated with increased risky behaviours,[3 4] can impact on the management of other chronic health conditions, such as adherence to HIV treatment[4–6] and can adversely affect educational and employment achievements.

Despite an increased focus on mental health programming and services, the global burden of mental health conditions is estimated to have remained relatively constant between 1990 and 2019.[1] In many low-income and middle-income settings, there are considerable treatment gaps including a lack of services and/or adequately trained staff.[7] Also, YP may not be aware of the available mental health services or find it difficult to

attend health facilities as they are at school or at work. Additional barriers include stigma around mental health conditions and YP's mistrust in service provision.

Understanding the epidemiology of mental health disorders in YP is essential to inform appropriate service provision and intervention development, and to inform policies and resource allocation. One of the challenges facing adolescent mental health programming for YP globally are gaps in data on the prevalence and determinants of disorders.[8 9] In Zimbabwe, there are limited population-level data on the mental health of YP. A 2006 population-based survey on mental health disorders among rural 15–23-year olds found 52% to be at risk of being affected and 24% to be at risk of being severely affected of common mental health disorders (CMDs).[3] A 2009–2011 survey found a prevalence of psychological distress of 4.5% among males and 8.2% of females aged 15–24 years.[4] More recent published data on adolescent mental health in Zimbabwe are primarily from studies in adolescents living with HIV. For example, in 2016/2017, 20.4% of adolescents living with HIV recruited into a trial at public clinics in two rural districts were at risk of CMDs.[10]

The primary aim of this study was to estimate the prevalence of CMDs among YP. Secondary aims were to estimate the prevalence of emotional and behavioural disorders and to explore factors associated with CMDs.

## METHODS

A cross-sectional population-based survey was conducted in 2018 among YP (13–24 years) in five communities in urban and peri-urban Harare and Mashonaland East, Zimbabwe. The communities were participating in a cluster randomised trial of a community-based, multicomponent HIV and sexual and reproductive health intervention for YP (the CHIEDZA trial). The aim of the survey was to describe YP's access to and use of technology including mobile phones, and to describe their mental health and well-being. The mobile phone and technology use results have been reported separately.[11] The survey was undertaken as part of formative research to inform the trial intervention, and the findings informed the development of the technology and general health information and counselling components of the intervention. The survey communities were purposively selected to represent both urban and peri-urban communities. The trial protocol has been published,[12] and randomisation and trial implementation occurred after this survey was completed.

Eligible participants were aged 13–24 years, resident in the study community at the time of the survey and either provided informed consent (16–24 years) or provided assent with guardian consent (age 13–15 years). A simple random sample of 100 GPS coordinates (primary sampling unit) were sampled per cluster from all potential points in the study areas using ArcGIS software V.10.5 (Esri, Redlands, USA). Points were randomly ordered

and then sequentially visited by a team of interviewers. All households with front doors within 20 m of the sampled GPS point were visited. The household head was interviewed to obtain basic demographic information about the household and was asked for consent to interview any eligible YP. If the household head was not available, another household member aged 16+ years or a neighbour was asked to provide information on the composition of the household to determine potential household eligibility. Households with YP were visited a further two times in order to interview the household head. All YP in the selected households were eligible for recruitment. The target sample size of 686 would provide ±6% precision around an expected prevalence of 50% or participants owning a mobile phone and assuming a design effect of 2 and 10% non-response.

### Data collection

Participants responded to a short (approximately 30 min) audio computer-assisted self-interviewing (ACASI) tablet-based questionnaire (online supplemental material). Questions were adapted from pre-existing questionnaires.[13–16] The questionnaire was developed in English and translated into Shona (the local language) and participants could respond in English or Shona. Modifications were made to the questionnaire following pretesting with the study team and following the pilot survey which was conducted outside the selected study sites.

The Shona Symptom Questionnaire (SSQ) is a locally validated 14-item scale asking about symptoms of CMDs in the past week with dichotomised yes/no response categories.[17] Two different versions of the SSQ were used. For piloting and the first 2 weeks of data collection (~20% of data collection), we used a Shona version that had been created by the study team through translation of the English version of the SSQ into Shona so that it would be understandable by YP. For the remainder of the survey, a self-completed version of the Shona SSQ which had previously been validated among 15–24-year olds was used.[18] A comparison of the two translation versions revealed no major but some slight differences. For example, the translation of 'My stomach was aching' was 'Ndairwadziwa nemudumbu' in version 1 (stomach was aching) and 'Pane pandaimborwadziwa nemudumbu' (sometimes had stomach ache) in version 2. Standard cut-offs of SSQ score ≥8 indicate risk of being affected by CMD and a score ≥11 indicate at risk of being severely affected by CMD.[17]

The Strengths and Difficulties Questionnaire (SDQ)[19] is a widely used freely available 25-item scale which measures emotional and behavioural problems in the past 6 months. A scoping review of the use of SDQ in Africa found 54 studies in 12 countries, however, concluded that very little was known about the psychometric properties of SDQ in African settings.[20] A validated Shona version of the SDQ does not exist. Shona speaking study team members created a Shona version which was then back translated and checked by a psychiatrist (DC). The SDQ has

five subscales: emotional symptoms, conduct problems, hyperactivity, peer problems and prosocial behaviour.[21] Each scale has five items each with the response categories ('not true', 'somewhat true' and 'certainly true') associated with a score between 0 and 2. The mean SDQ-25 score is used as a measure of psychosocial well-being, with the maximum total SDQ score (sum of all scales except the prosocial scale) of 40. In low-risk or general population samples, responses to SDQ items can be used to create a three-subscale division, that is, internalising problems (sum of emotional and peer scales, 10 items), externalising problems (sum of conduct and hyperactivity scales, 10 items) and the prosocial scale (five items).[22] Additionally, respondents can be described according to their SDQ 'caseness' profile: (1) neither subscale elevated, (2) elevated internalising subscale, (3) elevated externalising subscale and (4) both subscales elevated. To facilitate the use of SDQ as a screening tool, UK norms for mean total SDQ and subscale scores have been created to allow identification of the 80th–90th centile (borderline) and >90th centile (abnormal) scores.[19] We used the self-report version of the SDQ questionnaire for adolescents 11–17 years (https://www.sdqinfo.org/py/sdqinfo/b0.py).

## Data management and analysis

Data were collected and recorded using Open Data Kit (ODK) survey software with built-in logic checks and skip patterns on Android tablets. Data were analysed using STATA V.17.0 (StatCorp, TX, USA) using survey commands to account for the one-stage cluster sampling design.

The prevalence of CMDs was described according to the standard adult cut-offs of SSQ score ≥8 and score ≥11 defining those at risk of being affected and severely affected by CMDs, respectively. We also described the prevalence of those at risk of being affected by CMD using a more conservative cut-off of SSQ ≥9.[23] Cronbach's alpha intraclass correlation coefficient was calculated to assess internal consistency of the SSQ scale with high internal consistency traditionally defined as alpha>0.7. Adjusted Wald tests were used to compare proportions between subpopulations. Age-adjusted Wald tests were used to explore potential differences in the SSQ findings according to the translation version of the questionnaire used.

Mean SDQ scores (total, domain and subscale) and their internal consistencies were described. UK norms, and where necessary Zimbabwe-specific norms, were used to define normal, borderline and abnormal scores for each of the domains. We calculated the mean, SD and Chronbach's alpha for the SDQ scores according to sex. Age-adjusted differences in means according to sex were examined taking into account the clustered sampling design. The proportion scores in the 'normal', 'borderline' and 'abnormal' ranges in this Zimbabwean sample were calculated based on cut-offs derived by the SDQ authors using UK data.[24] As the use of UK norms have been shown to be inappropriate in other settings,[25] Zimbabwean cut-offs were generated to classify the 80th–90th centile and >90th centile bands. Participants were described according to their SDQ 'caseness' profile.

Participants with missing data for one or more of the SSQ or SDQ questions were excluded from the analysis of that scale. Multiple logistic regression was used to explore the factors associated with risk of being affected by CMDs (SSQ≥8), with adjustment for the one-stage cluster sampling design. Potential explanatory variables were age, sex, marital status, community of residence, highest level of school attended, current occupational status, religion, travel for at least 1 month in past 12 months, length of time living in the community and orphanhood status, with age being considered an a priori potential confounder. Wald tests adjusted for the clustered sampling design were used at each step of the analysis. Community D, where only 10 participants were interviewed due to logistical reasons, was excluded from regression analysis.

YP identified as needing further assessment based on their SSQ responses were referred immediately to health facilities for further assessment. Participants who scored 11 or higher on the SSQ or who responded 'yes' to either of the following questions were considered in need of further assessment: item 5 'In the past week, I sometimes saw or heard things which others could not see or hear'; item 11 'In the past week, at times I felt like committing suicide'. In such instances, the research assistant was required to accompany the respondent to the nearest health facility so that they could seek professional care for example, for mental health issues. At the time of study commencement, participants were referred to the 'Friendship Bench' service at local government health facilities. The Friendship Bench lay health workers deliver basic cognitive behavioural therapy with an emphasis on problem solving therapy, activity scheduling and peer led group support. However, local mental health services were reduced as staff were dealing with a cholera outbreak and YP were reluctant to attend as they were not comfortable discussing their issues with the elderly counsellors. An alternative referral pathway was established, whereby, YP were referred to a psychiatrist's private clinic. Travel and initial consultation expenses were covered by the project, and uptake of referrals was monitored through phone calls with the YP and through communication with the doctor.

## Patient and public involvement

Patients and the public were not directly involved in the design, conduct or analysis of this study. Formative work conducted in 2018 in the study communities as part of the CHIEDZA trial involved workshops and interviews with YP with the aim of understanding their health priorities and health service preferences.

## RESULTS

A total of 1212 households were sampled from 140 GPS point clusters in five communities (A 25, B 48, C 21, D

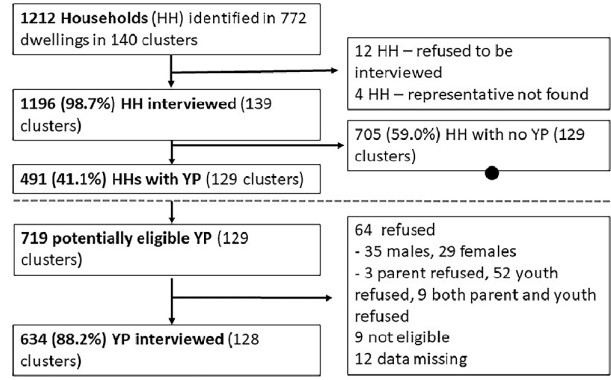

**Figure 1** Survey recruitment. HH, household; YP, young people.

42 and E 4) figure 1. YP in the target age range (n=719) were identified in 491 households (41.1% of successfully interviewed households). Stratified sampling led to the recruitment of approximately 60% of participants from Harare, and 40% of the participants from Mashonaland East.

### Demographic profile of study population

A total of 634/719 (88.2%) eligible YP, mean age 18.0 years (SD 3.3) and 62.6% (397/634) female, participated. The majority (83.9%, 532/634) had never been married, 86.8% (550/634) had attended secondary school or higher and only 14.7% (51/346) out of school participants reported that they were working. Over a quarter (28.7%, 182/634) reported that they had travelled for at least 1 month in the past 12 months and 17.7% (112/634) had lived less than 1 year in their community. Approximately one-third of respondents reported that one or both of their parents were dead or that a parent's location was unknown (table 1).

### Common mental health disorders

The median SSQ score was 7 (IQR 4, 9) overall, 6 (IQR 4, 9) in males and 7 (IQR 4, 9) in females. Cronbach's alpha was 0.77 suggesting that the instruments are measuring the same construct (mood). In total, 37.38% (237/634) were at risk of CMD (SSQ≥8) and 11.51% (73/634) were severely at risk of CMD (SSQ≥11). Using a more conservative cut-off of SSQ≥9, 28.55% (181/634) were at risk of CMD. A fifth of those interviewed (22.08%) were identified as needing further assessment, that is, had a SSQ≥11 or reported having seen or heard things which others could not see or hear (9.78% (62/634)) or that they felt like committing suicide in the past week 11.20% (71/634). The prevalence of CMD was similar according to the translation version of the questionnaire used (online supplemental table S1). The prevalence of CMD was similar among males and females. However, males were more likely than females to report having heard or seen things which others could not see or hear (14.35% vs 7.05%, p=0.01) and females were more likely to report suicidal ideation than males (13.60% vs 7.17%, p=0.01) (table 2). Individual SSQ question responses and

outcomes according to age group are presented in online supplemental table S2.

A total of 140 YP out of 634 (22.08%) were identified as needing immediate referral for assessment by a psychiatrist, however, we were only able to confirm that 23 of these YP (16%) were seen by the psychiatrist. The study team attempted to revisit the YP to encourage them to attend the referral but many refused, were too busy or had relocated. It is possible that some participants may have accessed the Friendship Bench at local health facilities.

### Prevalence of emotional and behavioural disorders

The mean total difficulties SDQ score was 10.7 (SD 6.3, alpha 0.75), externalising symptoms score was 4.31 (SD 3.48, alpha 0.68) and internalising symptom score was 6.43 (SD 3.94, alpha 0.63). The mean scores for emotional symptoms, conduct problems, hyperactivity/inattention and peer problems were 3.66, 2.22, 2.09 and 2.77, respectively. The mean prosocial behaviour score was 8.13 (online supplemental table S3).

Cronbach's alpha, an indication of internal consistency of the measure, was 0.75 for the total difficulties score and ranged from 0.38 to 0.68 for subscales. There was no evidence of a sex difference in total difficulties score and weak evidence of males having higher externalising scores than females (4.63 vs 4.11, p=0.05) and females having higher internalising scores than males (6.64 vs 6.08, p=0.08) (online supplemental table S4).

Using UK norms to define normal, borderline and abnormal scores for each of the domains, a high proportion (15.8%) of Zimbabwean YP had abnormal scores for emotional symptoms and a low proportion had abnormal scores for hyperactivity/inattention scores (2.8%) and prosocial scores (7.1%). Using the Zimbabwean study data, we created country-specific cut-offs where the scores within each band differed for the emotional symptoms, hyperactivity/attention and prosocial domains only. Using the Zimbabwean norms, a total of 13.9% had an elevated internalising subscale, 11.5% an elevated externalising subscale and 6.9% with both scales elevated (table 3).

### Correlates of being at risk of common mental health disorders

The odds of CMD increased by 9% with each year increase in age (OR 1.09, p<0.001). Those who were out of school and not working had higher odds of CMD compared with those in school or those working (adjusted OR 1.67 (1.07, 2.62), p=0.04). Following adjustment for age, there was no evidence of an association between risk of CMD and participants' sex, marital status, education level, religion, travel in the past 12 months, length lived in the community or orphanhood status (table 4).

## DISCUSSION
### Principal findings

We observed a high prevalence of self-reported symptoms of CMD in this population sample of 13–24-year olds in

**Table 1** Demographic characteristics of the study population (N=634)

| | Sex of respondent | | | | | |
| --- | --- | --- | --- | --- | --- | --- |
| | Male (n=237, 37.4%) | | Female (n=397, 62.6%) | | Total (n=634) | |
| | No. | % | No. | % | No. | % |
| Age group (years) | | | | | | |
| 13–15 years | 65 | 27.4 | 96 | 24.2 | 161 | 25.4 |
| 16–17 years | 47 | 19.8 | 74 | 18.6 | 121 | 19.1 |
| 18–19 years | 55 | 23.2 | 89 | 22.4 | 144 | 22.7 |
| 20–24 years | 70 | 29.5 | 138 | 34.8 | 208 | 32.8 |
| Mean age (years) | 17.8 (17.4, 18.2) | | 18.2 (17.8, 18.5) | | 18.0 (17.8, 18.3) | |
| Marital status | | | | | | |
| Married | 5 | 2.1 | 64 | 16.1 | 69 | 10.9 |
| Cohabiting | 1 | 0.4 | 15 | 3.8 | 16 | 2.5 |
| Never married | 225 | 94.9 | 307 | 77.3 | 532 | 83.9 |
| Divorced/separated | 6 | 2.5 | 11 | 2.8 | 17 | 2.7 |
| Highest level of school attended | | | | | | |
| Primary | 32 | 13.5 | 50 | 12.6 | 82 | 12.9 |
| Secondary | 195 | 82.3 | 330 | 83.1 | 525 | 82.8 |
| Higher (tertiary) | 9 | 3.8 | 16 | 4 | 25 | 3.9 |
| Never been to school | 1 | 0.4 | 1 | 0.3 | 2 | 0.3 |
| Current occupational status | | | | | | |
| In school/university | 121 | 51.1 | 167 | 42.1 | 288 | 45.4 |
| Out of school (working) | 22 | 9.3 | 29 | 7.3 | 51 | 8 |
| Out of school (not working) | 94 | 39.7 | 201 | 50.6 | 295 | 46.5 |
| Religion* | | | | | | |
| Roman Catholic | 27 | 11.5 | 38 | 9.6 | 65 | 10.3 |
| Protestant | 55 | 23.5 | 102 | 25.7 | 157 | 24.9 |
| Pentecostal | 96 | 41.0 | 169 | 42.6 | 265 | 42.0 |
| Apostolic | 19 | 8.1 | 67 | 16.9 | 86 | 13.6 |
| Other Christian/Muslim/Other | 5 | 2.1 | 3 | 0.8 | 8 | 1.3 |
| No religion | 32 | 13.7 | 18 | 4.5 | 50 | 7.9 |
| Travelled for at least 1 month in past 12 months | | | | | | |
| No | 170 | 71.7 | 282 | 71 | 452 | 71.3 |
| Yes | 67 | 28.3 | 115 | 29 | 182 | 28.7 |
| How long lived in community?† | | | | | | |
| <1 year | 29 | 12.3 | 83 | 20.9 | 112 | 17.7 |
| 1–4 years | 44 | 18.7 | 98 | 24.7 | 142 | 22.5 |
| 5+ years | 162 | 68.9 | 216 | 54.4 | 378 | 59.8 |
| Orphan status | | | | | | |
| Double orphan | 23 | 9.7 | 40 | 10.1 | 63 | 9.9 |
| Mother dead, father alive | 21 | 8.9 | 31 | 7.8 | 52 | 8.2 |
| Mother alive, father dead or unknown | 40 | 16.9 | 70 | 17.6 | 110 | 17.4 |
| Both parents alive | 153 | 64.6 | 256 | 64.5 | 409 | 64.5 |

*n=3 no response.
†n=2 do not know.

**Table 2** Shona Symptom Questionnaire (SSQ): item response rates by sex and overall response (n=634)

| | Total* | Male* | Female* | P value for age-adjusted difference between sexes† |
|---|---|---|---|---|
| n | 634 | 237 | 397 | |
| Median SSQ (IQR) | 7 (4 to 9) | 6 (4 to 9) | 7 (4 to 9) | |
| Mean SSQ (95% CI) | 6.44 (6.13 to 6.74) | 6.37 (5.95 to 6.79) | 6.48 (6.11 to 6.84) | p=0.811 |
| At risk for CMD (SSQ≥8) | 37.38 (32.99 to 41.99) | 34.18 (27.76 to 41.23) | 39.29 (34.11 to 44.73) | p=0.278 |
| At risk for CMD (SSQ≥9) | 28.55 (24.88 to 32.53) | 26.58 (21.49 to 32.39) | 29.72 (25.22 to 34.65) | p=0.443 |
| Severely at risk of CMD (SSQ≥11) | 11.51 (9.08 to 14.51) | 10.97 (7.86 to15.11) | 11.84 (8.79 to 15.76) | p=0.803 |
| I sometimes saw or heard things which others could not see or hear | 9.78 (7.47 to 12.71) | 14.35 (10.03 to 20.11) | 7.05 (4.72 to 10.41) | p=0.011 |
| At times I felt like committing suicide | 11.20 (9.01 to 13.84) | 7.17 (4.57 to 11.09) | 13.60 (10.69 to 17.16) | p=0.012 |
| In need of further assessment | 22.08 (18.55 to 26.07) | 21.10 (16.20 to 27.00) | 22.67 (18.36 to 27.66) | p=0.693 |

*Mean and proportions taking into account clustered study design but not adjusted for age.
†Adjusted Wald test accounting for clustered study design and adjusting for age in years.
CMD, common mental health disorders.

urban and peri-urban Zimbabwe. Over one in five YP required immediate referral for psychiatrist assessment, but there was very low uptake of referral to mental health services. The odds of being affected by CMD increased with age and was higher among those out of school and not working. Self-reported emotional and behavioural disorders were also common. Compared with UK norms for the SDQ, a lower proportion had abnormal scores for hyperactivity/inattention and prosocial subscales and a higher proportion had abnormal emotional symptoms scores which necessitated the creation of Zimbabwe-specific norms.

### Common mental and emotional and behavioural disorders
We found 37% of YP to be at risk of CMD (SSQ≥8), and 12% to be severely at risk of CMD (SSQ≥11). High prevalence of symptoms of CMD was also observed in a 2006 population-based survey among rural Zimbabwean youth (15–23 years),[3] and among HIV+ adolescents in Harare in 2009[5] and in two rural districts in 2016/2017.[10] Considerably lower prevalence of symptoms of CMDs was observed in a 2009–2011 general population survey of 15–24 year olds.[4 26] While all these studies used the SSQ, it is difficult to make comparisons given the different populations, socioeconomic environments and modes of data collection used, that is, self-completed versus interviewer administered. Previous research has shown that symptom reporting varied according to method with higher reporting in studies using self-administered methods.[27] The observed increase in risk of CMD with age is consistent with the literature.[28–31] We did not observe a higher prevalence of CMD among females compared with males which is in contrast to other adolescent studies.[32 33]

Similar high prevalence of CMD has been observed in systematic reviews of studies among adolescents in sub-Saharan Africa with estimated prevalence of 27% for depression and 30% for anxiety disorders,[28] and 30–50% for emotional or behavioural difficulties or significant psychological distress in adolescents with HIV.[29]

The high prevalence of symptoms of CMD among YP in Zimbabwe may in part be due to the challenging economic situation with limited job opportunities, and the HIV and associated orphanhood epidemic.[34] Higher prevalence of CMD symptoms among those out of school and not working in this study has also been observed in HIV+ adolescent populations.[29] Interviews with adolescents who participated in the recent youth version of the Friendship Bench suggest that hopelessness and a feeling of lack of control over one's situation have an important impact on mental well-being.[35]

### Measurement of common mental health and emotional and behavioural disorders
This study demonstrated that self-reported SSQ and SDQ are feasible to measure in 13–24-year olds in urban and peri-urban Zimbabwe. While the SSQ has been validated in several populations,[17 23] there is no consensus on the most appropriate cut-offs to be used for YP.[3 10 26] Some authors have found that a cut-off of ≥5 provides the best performance and increases sensitivity of the screening tool,[26] however, their conclusions were based on interviewer-administered questionnaires where some of the SSQ questions had non-standard wording. A more recent validation of SSQ against MINI-KID in 12–17-year olds attending primary care clinics in urban Zimbabwe

**Table 3** Strength and Difficulties Questionnaire (SDQ): banding of self-report raw scores in Zimbabwe and UK

| | | UK norms | | | | Zimbabwe norms | | | |
| --- | --- | --- | --- | --- | --- | --- | --- | --- | --- |
| | | | All (n=634) | Male (n=237) | Female (n=397) | | All (n=634) | Male (n=237) | Female (n=397) |
| | | Band | % | % | % | Band | % | % | % |
| Total difficulties | Normal | 0–15 | 78.4 | 78.9 | 78.1 | 0–15 | 78.4 | 78.9 | 78.1 |
| | Borderline | 16–19 | 11 | 11.4 | 10.8 | 16–19 | 11.0 | 11.4 | 10.8 |
| | Abnormal | 20–40 | 10.6 | 9.7 | 11.1 | 20–40 | 10.6 | 9.7 | 11.1 |
| Emotional symptoms | Normal | 0–5 | 74.1 | 77.6 | 72.0 | 0–5 | 74.1 | 77.6 | 72.0 |
| | Borderline | 6 | 10.1 | 10.1 | 10.1 | 6–7 | 13.7 | 13.9 | 13.6 |
| | Abnormal | 7 to 10 | 15.8 | 12.2 | 17.9 | 8–10 | 12.2 | 8.4 | 14.4 |
| Conduct problems | Normal | 0–3 | 74.9 | 75.1 | 74.8 | 0–3 | 74.9 | 75.1 | 74.8 |
| | Borderline | 4 | 12.8 | 11.0 | 13.9 | 4 | 12.8 | 11.0 | 13.9 |
| | Abnormal | 5 to 10 | 12.3 | 13.9 | 11.3 | 5–10 | 12.3 | 13.9 | 11.3 |
| Hyperactivity/ inattention | Normal | 0–5 | 92.3 | 89.9 | 93.7 | 0–3 | 77.3 | 72.6 | 80.1 |
| | Borderline | 6 | 4.9 | 6.8 | 3.8 | 4 | 11.0 | 13.5 | 9.6 |
| | Abnormal | 7–10 | 2.8 | 3.4 | 2.5 | 5–10 | 11.7 | 13.9 | 10.3 |
| Peer problems | Normal | 0–3 | 64.7 | 66.2 | 63.7 | 0–3 | 64.7 | 66.2 | 63.7 |
| | Borderline | 4–5 | 22.7 | 23.2 | 22.4 | 4–5 | 22.7 | 23.2 | 22.4 |
| | Abnormal | 6–10 | 12.6 | 10.6 | 13.9 | 6–10 | 12.6 | 10.6 | 13.9 |
| Prosocial behaviour | Normal | 6–10 | 89.9 | 90.7 | 89.4 | 7–10 | 79.2 | 77.6 | 80.1 |
| | Borderline | 5 | 3.0 | 3.4 | 2.8 | 6 | 10.7 | 13.1 | 9.3 |
| | Abnormal | 0–4 | 7.1 | 5.9 | 7.8 | 0–5 | 10.1 | 9.3 | 10.6 |
| SDQ caseness profile | Neither subscale elevated | | | | | | 67.7 | 67.9 | 67.5 |
| | Elevated internalising subscale | | | | | | 13.9 | 10.1 | 16.1 |
| | Elevated externalising subscale | | | | | | 11.5 | 15.6 | 9.1 |
| | Elevated internalising and externalising subscale | | | | | | 6.9 | 6.3 | 7.3 |

found the optimal cutpoint of SSQ≥8 (Unpublished data).

In this first documented use of a Shona language version of the SDQ, we observed good internal consistency for total difficulties but lower consistency within some of the subscales. In line with studies in other sub-Saharan African settings, the translation of a few of the terms within the SDQ were challenging,[20] for example, in the hyperactivity subscale, there is a question about 'fidgeting' for which there is no equivalent term in Shona. The patterns of emotional and behavioural disorders observed in this study (lower abnormal prosocial and hyperactivity subscale scores, higher emotional and peer subscale scores) are similar to patterns observed in studies of South African adolescents[25] and children[36] and Namibian adolescents.[37]

Multiple screening tools exist for CMDs and while the SSQ is widely used in Zimbabwe, and the SDQ used widely globally, there may be other tools that are more appropriate for the Zimbabwean adolescent population. A first

step will therefore be to decide on which tool or tools are most promising and then plan the necessary careful adaptation and validation work. UNICEF with WHO and other partners are currently developing tools for the Measurement of Mental Health Among Adolescents at the Population Level with a focus on Patient Health Questionnaire (PHQ-9) and Generalized Anxiety Disorder Assessment (GAD-7).[38] If the SDQ is to be used again in Zimbabwean adolescents, then additional work is needed to develop a Shona version and then to develop country-specific SDQ cut-offs for 'caseness'.[25]

### Services for young people

The high burden of CMD among YP and reluctance to attend adult mental health services demonstrate an urgent need for youth-friendly mental health services. The original referral pathway was to refer the YP to the Friendship Bench Service at nearby local government health facilities. However, health facility staff shortages due to an ongoing cholera outbreak coupled with YPs

**Table 4** Associations between demographic characteristics and ≥8 score on SSQ

| | N | N | % | Crude OR | Age adj. OR |
|---|---|---|---|---|---|
| **Sex** | | | | | |
| Male | 237 | 81 | 34.18 (27.76, 41.23) | REF | REF |
| Female | 387 | 150 | 38.76 (33.61, 44.18) | 1.22 (0.87, 1.72) | 1.19 (0.84, 1.67) |
| | | | p=0.254* | p=0.254 | |
| **Age group** | | | | | |
| 13–15 years | 159 | 42 | 26.42 (19.46, 34.79) | REF | |
| 16–17 years | 119 | 47 | 39.50 (31.17, 48.48) | 1.86 (1.14, 3.05) | |
| 18–19 years | 144 | 58 | 40.28 (31.98, 49.18) | 1.91 (1.16, 3.14) | |
| 20–24 years | 202 | 84 | 41.58 (35.67, 47.75) | 2.12 (1.37, 3.27) | |
| | | | p=0.0117 | | |
| **Age in years** | | | | 1.09 (1.04, 1.15) | |
| | | | | p<0.001 | |
| **Marital status** | | | | | |
| Married/cohabiting | 79 | 40 | 50.63 (40.54, 60.68) | REF | REF |
| Never married | 528 | 182 | 34.47 (29.76, 39.51) | 0.51 (0.32, 0.82) | 0.68 (0.40, 1.13) |
| Widowed/divorced/separated | 17 | 9 | 52.94 (31.08, 73.73) | 1.10 (0.37, 3.23) | 1.06 (0.36, 3.17) |
| | | | p=0.0071 | p=0.0035 | p=0.22 |
| **Highest level of school attended** | | | | | |
| None/primary | 82 | 21 | 25.61 (16.10, 38.19) | REF | REF |
| Secondary | 517 | 203 | 39.26 (34.37, 44.38) | 1.88 (1.01, 3.51) | 1.32 (0.67, 2.60) |
| Higher | 25 | 7 | 28.00 (13.96, 48.25) | 1.13 (0.43, 2.94) | 0.57 (0.21, 1.56) |
| | | | p=0.0696 | p=0.1228 | p=0.1625 |
| **Occupation** | | | | | |
| In school | 287 | 83 | 28.92 (23.13, 35.50) | REF | REF |
| Out of school working | 51 | 18 | 35.29 (22.70, 50.33) | 1.34 (0.71, 2.52) | 0.99 (0.49, 2.00) |
| Out of school not working | 286 | 130 | 45.45 (39.45, 51.60) | 2.05 (1.40, 2.99) | 1.67 (1.07, 2.62) |
| | | | p=0.0006 | p=0.0013 | p=0.0410 |
| **Religion** | | | | | |
| Roman Catholic | 64 | 22 | 34.38 (23.59, 47.05) | REF | REF |
| Other Christian/Muslim/no religion | 557 | 208 | 37.34 (32.72, 42.21) | 1.14 (0.65, 1.98) | 1.19 (0.67, 2.11) |
| | | | p=0.6452 | p=0.6454 | p=0.5538 |
| **Community of residence** | | | | | |
| A | 178 | 71 | 39.89 (31.41, 49.02) | REF | REF |
| B | 140 | 48 | 34.29 (26.11, 43.51) | 0.78 (0.46, 1.35) | 0.78 (0.45, 1.34) |
| C | 147 | 64 | 43.54 (34.60, 52.91) | 1.16 (0.69, 1.97) | 1.16 (0.68, 1.97) |
| E | 159 | 48 | 30.19 (22.73, 38.87) | 0.65 (0.38, 1.11) | 0.66 (0.38, 1.13) |
| | | | | p=0.1625 | p=0.1711 |
| **Travelled for at last 1 month in last 12 months** | | | | | |
| No | 444 | 161 | 36.26 (31.79, 40.99) | REF | REF |
| Yes | 180 | 70 | 38.89 (30.47, 48.03) | 1.12 (0.76, 1.65) | 1.11 (0.75, 1.65) |
| | | | p=0.5705 | p=0.5706 | p=0.5980 |
| **How long lived in the community** | | | | | |
| <1 year | 112 | 42 | 37.50 (29.04, 46.80) | REF | REF |
| 1–4 years | 138 | 56 | 40.58 (32.45, 49.26) | 1.14 (0.72, 1.81) | 1.14 (0.70, 1.84) |

Continued

**Table 4** Continued

| | N | N | % | Crude OR | Age adj. OR |
|---|---|---|---|---|---|
| 5+ years | 372 | 133 | 35.75 (30.37, 41.52) | 0.93 (0.60, 1.44) | 0.99 (0.64, 1.54) |
| | | | p=0.5865 | p=0.6234 | p=0.7846 |
| Orphan status | | | | | |
| Double orphan | 61 | 28 | 45.90 (33.22, 59.14) | REF | REF |
| Mother dead, father alive | 52 | 25 | 48.08 (35.92, 60.46) | 1.09 (0.51, 2.33) | 1.08 (0.51, 2.31) |
| Mother alive, father dead | 108 | 36 | 33.33 (24.94, 42.93) | 0.59 (0.30, 1.15) | 0.61 (0.31, 1.20) |
| Both parents alive | 403 | 142 | 35.24 (29.83, 41.04) | 0.64 (0.37, 1.11) | 0.76 (0.44, 1.31) |
| | | | p=0.1118 | p=0.0861 | p=0.2217 |

Community D (n=10) dropped from this analysis so total N is 624.
Adjusted for age in years.
*P value from adjusted wald test.
SSQ, Shona Symptoms Questionnaire.

reluctance to access adult Friendship Bench services required the creation of an alternative referral pathway. While some YP successfully attended referral appointments at the hospital in Harare, there were many YP who did not attend care. This study has highlighted the need to involve YP when developing research study protocols and to establish the feasibility and acceptability of proposed referral pathways. In addition to increasing the availability of high-quality youth-friendly mental health services, increased education and skill-building for YP and their families may help to support healthy mental health and well-being.[31]

### Strengths and weaknesses of the study

A strength of this study is a representative general population sample of YP in five urban and peri-urban communities in Zimbabwe, in the context of limited population-level estimates of mental health disorders among YP. We demonstrated the feasibility of using ACASI to collect self-reported measures of mental health and emotional and behavioural disorders in this young population. Only the self-report version of the SDQ was used, however, for optimal use, it is recommended that teachers and parents also complete the SDQ allowing triangulation of data.[39] The tools used were not validated for use in the full age group, the translation of the SDQ into Shona did not follow recommended procedures, and two different Shona versions of SSQ were used. We do not think that the use of different SSQ versions had a big impact on the results, for example, similar mean and median SSQ scores (online supplemental table S1). When version 1 was used, there was a higher prevalence of those at risk for CMD compared with when version 2 was used, but the differences may have been due to population characteristics and not the tools themselves. The self-reported measures used may have been subject to recall bias and social desirability bias.

### CONCLUSION

We observed a high prevalence of probable CMDs among general population urban and peri-urban YP, however, there was a low uptake of referrals to existing mental health services. The use of self-reported SSQ and SDQ was feasible in this population but additional research is needed to determine the validity and most appropriate cut-offs for these measurement tools in young Zimbabweans. Important outstanding research questions for YP's mental health in this resource-constrained setting include how to identify those most at risk, and how to ensure that they are linked to appropriate youth-focused mental health services.

**Author affiliations**
[1]Medical Research Council International Statistics and Epidemiology Group, Faculty of Epidemiology and Population Health, London School of Hygiene & Tropical Medicine, London, UK
[2]The Health Research Unit Zimbabwe (THRU ZIM), Biomedical Research and Training Institute, Harare, Zimbabwe
[3]Department of Psychiatry, College of Health Sciences University of Zimbabwe, Harare, Zimbabwe
[4]Centre for Global Mental Health, London School of Hygiene & Tropical Medicine, London, UK
[5]Department of Clinical Research, London School of Hygiene & Tropical Medicine, London, UK

**Acknowledgements** We thank the study participants and their communities for participating and facilitating this research. We are grateful to the field team who conducted the interviews, and the other Biomedical Research and Training institute, Harare, who supported the study.

**Contributors** Obtained funding: AMD and RF; responsible for overall content as guarantor: AMD and RF; wrote the study protocol: AMD, TB, ED, GM, CG, DC and RF; assisted with fieldwork and data collection which was overseen by: AMD, ED and TB; did central data monitoring: TB; conducted statistical analysis: AMD; advised on statistical analysis: VS; drafted the initial manuscript: AMD; provided substantial edits to the manuscript: TB, ED, GM, CG, VS, DC and RF; all authors contributed to interpretation of the results and have read and approved the final manuscript.

**Funding** Financial support for this study was provided through joint funding under the UK Medical Research Council (MRC)/UK Department for International Development (DFID) Concordat agreement which is supported by the European Union under the EDCTP2 programme (reference MR/K012126/1) with AMD

receiving support (G0700837). AMD is supported by a UKRI Future Leader Fellowship (grant number MR/T043156/1). RF is funded by the Wellcome Trust through a Senior Fellowship in Clinical Science (206316Z/17/Z).

**Competing interests**  None declared.

**Patient and public involvement**  Patients and/or the public were not involved in the design, or conduct, or reporting or dissemination plans of this research.

**Patient consent for publication**  Not required.

**Ethics approval**  This study involves human participants and was approved by Biomedical Research and Training Institute (BRTI, AP149/2018), Medical Research Council of Zimbabwe (MCRZ, MRCZ/A/2362), London School of Hygiene and Tropical Medicine Research Ethics Committee (LSHTM REC, No.15919). Participants gave informed consent to participate in the study before taking part.

**Provenance and peer review**  Not commissioned; externally peer reviewed.

**Data availability statement**  Data are available upon reasonable request. Data will be resposited in LSHTM Data Compass in March 2023.

**ORCID iDs**
Aoife Margaret Doyle http://orcid.org/0000-0002-3305-7738
Victoria Simms http://orcid.org/0000-0002-4897-458X

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
