## [Reviewer comments · BMJ Open]

ARTICLE DETAILS

TITLE (PROVISIONAL)	Common mental health and emotional and behavioural disorders among adolescents and young adults in Harare and Mashonaland East, Zimbabwe: A population-based prevalence study
AUTHORS	Doyle, Aoife; Bandason, T; Dauya, E; McHugh, Grace; Grundy, Chris; Simms, Victoria; Chibanda, D; Ferrand, Rashida

VERSION 1 – REVIEW

REVIEWER	Bhavsar, Vishal Institute of Psychiatry Department of Health Service and Population Research, Department of Health Services and Population Research
REVIEW RETURNED	22-Aug-2022

GENERAL COMMENTS	Well done. I wonder if the terminology "mental health illness" is tautological or unclear- I would support something like common mental disorders, mental health conditions etc. I would consider splitting the first sentence of the Methods into two shorter sentences. In the second para of Data Collection, I would clarify symptoms of what.
---

REVIEWER	Loades, Maria Bristol Medical School, Department of Psychology
REVIEW RETURNED	01-Dec-2022

GENERAL COMMENTS	Overall, this is a well reported and nicely conducted study, that adds to the literature on the mental health burden in youth in LMIC contexts. I have a few small suggestions which I believe would improve the manuscript pre-publication: 1) describe in more detail the differences between the 2 versions of the SSQ used (method section), and in the discussion section, explain how you think this may have influenced the results 2) comment on whether you think the study was sufficiently powered, given the assumption of 50% prevalence, and the finding of <40% prevalence 3) be careful throughout that the language used doesn't objectify young people - e.g. young people are not 'red flags' - although their scores on questionnaires might exceed thresholds that indicate red flag status, and young people are not 'abnormal' - even if their scores on the SDQ fall within abnormal range or are classified as abnormal. 4) when reporting Cronbach's alpha (p8, results section), please reword to make sure it is clear that this is a measure of internal consistency of a measure, Not a measure of validity - i.e. it indicates that the scores on items were consistent with one another, but not that these necessarily reflect the construct of mood. 5) the SDQ is better described throughout as a measure of emotional and behavioural difficulties, not just 'behavioural difficulties'.
--

VERSION 1 – AUTHOR RESPONSE

Reviewer: 1

Dr. Vishal Bhavsar, Institute of Psychiatry Department of Health Service and Population Research

Comments to the Author:

Well done.

Response: Thanks for the feedback and helpful suggestions.

I wonder if the terminology "mental health illness" is tautological or unclear- I would support something like common mental disorders, mental health conditions etc.

Response: We have changed this to 'mental health conditions.

I would consider splitting the first sentence of the Methods into two shorter sentences.

Response: We have made this change.

In the second para of Data Collection, I would clarify symptoms of what.

Response: We have clarified that these were symptoms of common mental health disorders.

Reviewer: 2

Dr. Maria Loades, Bristol Medical School, University of Bath

Comments to the Author:

Overall, this is a well reported and nicely conducted study, that adds to the literature on the mental health burden in youth in LMIC contexts.

Response: Thanks for reviewing the paper and for your helpful suggestions.

I have a few small suggestions which I believe would improve the manuscript pre-publication:

1) describe in more detail the differences between the 2 versions of the SSQ used (method section), and in the discussion section, explain how you think this may have influenced the results

Response: The native Shona speakers in our team reviewed the two versions of the SSQ and noted some slight differences in some of the translations. We have added the following text to the methods section:

'A comparison of the two translation versions revealed no major but some slight differences. For example, the translation of 'My stomach was aching' was 'Ndairwadziwa nemudumbu' in version 1 (stomach was aching) and 'Pane pandaimborwadziwa nemudumbu' (sometimes had stomach ache) in version 2.'

The results according to translation version are presented in Table S1 in the supplementary material. We do not think that the version differences had a big impact on the results. In the discussion we have added the following text:

'We do not think that the use of different SSQ versions had a big impact on the results e.g. similar mean and median SSQ scores (Table S1). When version 1 was used, there was a higher prevalence of those at risk for CMD compared to when version 2 was used, but the differences may have been due to population characteristics and not the tools themselves.'

2) comment on whether you think the study was sufficiently powered, given the assumption of 50% prevalence, and the finding of <40% prevalence

Response: The target sample size of 686 would provide +/-6% precision around an expected prevalence of 50% and assuming a design effect of 2 and 10% non-response. The observed lower prevalence of 37.4% gave us a greater precision on this estimate i.e. the confidence interval is 33.0 to 42.0 so a precision of +/-4%. This increased precision occurs as a proportion of 50% indicates the maximum variability in a population so a higher or lower proportion is associated with lower variability.

3) be careful throughout that the language used doesn't objectify young people - e.g. young people are not 'red flags' - although their scores on questionnaires might exceed thresholds that indicate red flag status, and young people are not 'abnormal' - even if their scores on the SDQ fall within abnormal range or are classified as abnormal.

Response: Thanks- this is a good point. We have updated the language to replace 'red flag' with YP identified as needing further assessment.

4) when reporting Cronbach's alpha (p8, results section), please reword to make sure it is clear that this is a measure of internal consistency of a measure, Not a measure of validity - i.e. it indicates that the scores on items were consistent with one another, but not that these necessarily reflect the construct of mood.

Response: We have edited the text as follows to make this important point clearer

` The Cronbach's alpha, an indication of internal consistency of the measure, was 0.75 for the total difficulties score and ranged from 0.38 to 0.68 for sub-scales.'

5) the SDQ is better described throughout as a measure of emotional and behavioural difficulties, not just 'behavioural difficulties'.

Response: Thanks. We have updated this throughout the text.